# Of Money and Men: A Scoping Review to Map Gender Barriers to Immunization Coverage in Low- and Middle-Income Countries

**DOI:** 10.3390/vaccines12060625

**Published:** 2024-06-05

**Authors:** Anna Kalbarczyk, Natasha Brownlee, Elizabeth Katz

**Affiliations:** 1Department of International Health, Johns Hopkins Bloomberg School of Public Health, Baltimore, MD 21205, USA; 2Global Center for Gender Equality, Washington, DC 20036, USA; natasha.brownlee@gcfge.org (N.B.); egkatz@usfca.edu (E.K.); 3Department of Economics, University of San Francisco, San Francisco, CA 94117, USA

**Keywords:** gender, immunization coverage, vaccination, LMICs, inequality

## Abstract

Among the multiple factors impeding equitable childhood immunization coverage in low- and middle-income countries (LMICs), gender barriers stand out as perhaps the most universal. Despite increasing recognition of the importance of gender considerations in immunization programming, there has not yet been a systematic assessment of the evidence on gender barriers to immunization. We conducted a scoping review to fill that gap, identifying 92 articles that described gender barriers to immunization. Studies documented a range of gender influencers across 43 countries in Africa and South Asia. The barrier to immunization coverage most frequently cited in the literature is women’s lack of autonomous decision-making. Access to immunization is significantly impacted by women’s time poverty; direct costs are also a barrier, particularly when female caregivers rely on family members to cover costs. Challenges with clinic readiness compound female caregiver’s time constraints. Some of the most important gender barriers lie outside of the usual purview of immunization programming but other barriers can be addressed with adaptations to vaccination programming. We can only know how important these barriers are with more research that measures the impact of programming on gender barriers to immunization coverage.

## 1. Introduction

Inequality in immunization is normally interpreted to refer to discrepancies in coverage by socioeconomic status, demographic characteristics, and/or geography [1,2]. However, these factors should be understood not only as gradients of vaccine coverage but also as drivers of effective demand for immunization. In other words, inequality is both a cause and a consequence with respect to vaccination.

Among the multiple factors impeding equitable childhood immunization coverage in low- and middle-income countries (LMICs), gender barriers—defined as the ways in which gender roles, norms, and relations impede immunization program performance—stand out as perhaps the most universal [3,4]. This is in large part because, across LMIC geographies, while mothers bear primary responsibility for their children’s health, they often lack the resources and/or decision-making authority to access vaccination services [5]. Deepening our understanding of how these barriers operate, and which ones are most salient and pervasive, is key to designing interventions to effectively address them.

But despite increasing recognition of the importance of gender considerations in immunization programming—as evidenced, for example, by the recent WHO publication *Why Gender Matters: Immunization Agenda 2030*—there has not yet been a systematic assessment of the evidence on gender barriers to immunization [6]. This scoping review, funded by the Bill and Melinda Gates Foundation, seeks to fill that gap by documenting the findings of over 90 peer-reviewed research papers and analyzing the results within a coherent conceptual framework. Taken together, the evidence described in this review makes a compelling case that failing to address the significant gender barriers to immunization, particularly routine immunization, will impede progress towards achieving greater equity in vaccine coverage, especially in LMICs.

## 2. Methods

The research protocol followed the Preferred Items for Systematic Reviews and Meta-analysis extension for Scoping Reviews (PRISMA-ScR) guidelines but was not registered. To identify the published literature on gender barriers to immunization in SSA and South Asia, we conducted a search of three databases, PubMed, Embase, and CINAHL, on 26 September 2023. The search strategy was designed to capture the literature related to the three specific concepts, (1) gender, (2) immunization, and (3) geography (specifically Sub-Saharan Africa (SSA) and South Asia). A complete list of search terms can be found in the Appendix A. 

Results from the three databases were downloaded, combined, and deduplicated in Endnote. Unique files were then uploaded to Covidence (www.covidence.org), an online, collaborative scoping review management software. Articles were included if they captured all three concepts and were published in English between 2000 and the date of search. Articles were excluded if they did not meet these inclusion criteria, only discussed vaccine product development, and if the gender analysis was limited to sex disaggregation only.

Title and abstract reviews and full-text reviews followed the same procedures. Two independent reviewers reviewed each and conflicts were resolved by a third, independent reviewer or by team discussion. Systematic and scoping reviews were included so that the references included in them could be reviewed. Additional articles shared by experts were also included and underwent review.

Data were extracted in Covidence and analyzed in Microsoft Excel (https://www.microsoft.com/zh-cn/). The extraction form included questions about the country/region of study, disease/immunization of focus, target population, study design, gender barriers and their relevance along the immunization value chain, gender barrier analysis domains, and other key observations made by reviewers. Descriptive statistics were generated where data could be quantified. Data were initially charted along the immunization value chain, using a gender analysis matrix. This approach seeks to organize data along gender analysis domains (i.e., access to resources; distribution of labor, practices, and roles; norms, values, and beliefs; decision-making power and autonomy; and policies, laws, and institutions). Some manuscripts explicitly named gender barriers using these domains while others were less explicit. The team used their understanding of the influence of gender on health to categorize barriers when needed. Results largely converged in select domains (i.e., decision-making power), requiring an additional lens.

We then used Phillips et al.’s conceptual framework for effective vaccine coverage in LMICs to organize our findings by three principal determinants: (1) Intent to vaccinate, (2) Community Access, and (3) Health Facility Readiness [7]. This framework is specifically designed to be applicable in LMICs and represents a synthesis of multiple previous frameworks, accounting for the nuanced interactions between supply and demand side factors. Gender influencers which had initially been organized using the gender analysis matrix were then grouped into the Intent, Access, or Readiness categories.

## 3. Results

A total of 3390 references were added to Endnote (version X9), a reference management software, and 845 duplicates were removed. A total of 2545 references were screened during the title and abstract review and an additional 36 duplicates were manually identified during this process; 2359 studies did not meet inclusion criteria. A total of 173 articles were assessed during the full-text review and 63 studies were excluded because there was no full text (n = 9), the article was retracted (n = 1), no focus on gender (n = 36), not related to the immunization value chain (n = 14), analysis was limited to sex disaggregation (n = 2), or was not original research (n = 1). Seven systematic/scoping reviews were also reviewed to identify relevant references; reviews themselves were not included but did support the identification and inclusion of 17 additional references. In total, 92 articles were included in the final analysis which documented gender barriers to immunization. The PRISMA diagram documenting these findings is presented in Figure 1.

Most studies assessed barriers to childhood immunizations (n = 63). This was followed by HPV (n = 21), COVID-19 (n = 7), polio (n = 4), maternal vaccines (n = 2), yellow fever (n = 1), and H1N1 (n = 1).

Studies (focused on 25 or fewer countries) documented a range of gender influences of vaccination across 43 countries in Africa and South Asia (see Figure 2). The most frequently studied geographies included Nigeria (n = 21), Ethiopia (n = 12), and Pakistan (n = 10). Three large multi-country studies each contained data from approximately 160 countries. Many studies explored geographic pockets within a country in recognition of the additional and/or specific challenges faced in these unique contexts.

To assess gender barriers to immunization, studies used quantitative only methods (n = 46), qualitative only methods (n = 39), and mixed methods (n = 8). Of those using quantitative methods, 26 conducted secondary analyses on country Demographic Health Surveys (DHSs). Measurement of immunization coverage varied as well including comparisons between full vs. partial vs. zero dose and hesitancy, readiness, acceptance, and awareness. Measures of gender inequity varied even more widely; nine quantitative studies used gender scores including SWPER Global [8,9], Gender Inequality Ratio (GIR) [10], Gender Development Index (GDI) [11], Gender Inequality Index (GII) [11], and proxies/indexes for empowerment, decision-making power, and autonomy [12,13,14,15,16].

Along the immunization value chain, most studies documented barriers related to demand for immunization and local-level vaccine delivery; very few assessed those barriers influenced by cross-cutting market dynamics such as the supply chain or representation in leadership (see Table 1). Results from the gender barriers analysis are presented by Intent, Access, and Readiness domains, recognizing overlap and intersections. 

**Intent** to vaccinate includes attitudes, perceived norms, and perceived control and represents the demand for vaccines that would result in vaccination in the absence of other barriers.

The most common gender influence on intention is women’s **lack of autonomous decision-making** about their health and the health of their children (n = 45) [8,12,17,18,19,20,21,22,23,24,25,26,27,28,29,30,31,32,33,34,35,36,37,38,39,40,41,42,43,44,45,46,47,48,49,50,51,52,53,54,55,56,57,58,59]. Across geographies, many women rely on their husband’s or an elder’s permission to seek healthcare services, including immunization; this was found across Africa (n = 37) and South Asia (n = 16) (note that some studies include more than one country/region). In one study in Nigeria, women with high household decision-making were more likely to have their child fully immunized than women with low decision-making (OR = 1.64, CI: 1.25, 2.14, *p* < 0.001) [48]. Another study in Ethiopia found that mothers who made healthcare decisions jointly with their husbands were 1.88 times (95% CI [1.06–3.34]) more likely to vaccinate their children fully than when decisions were made by the husbands alone. Further, mothers who made healthcare decisions themselves were 4.03 (AOR 95% CI [1.66–9.78]) times more likely to fully vaccinate their children than when decisions were made by husbands alone [49]. Similar results were reported in Bangladesh, where women with more autonomy in healthcare decisions were more likely to have children who were fully vaccinated (86.1%) than those without autonomy (78.8%) (95% CI 1.079–2.317) [51]. Another study in South Africa found that children whose parents were both involved in shared decision making were less likely to have missed opportunities for vaccination than those whose immunization decision-making was not shared (OR = 0.21, 95% CI: 0.07–0.62) [52].

Other decision makers play important roles as well, sometimes interacting or compounding. For example, mothers-in-law were found to have an important influencing effect (n = 4) in Guinea, India, Nigeria, and Uganda [35,36,38,40]. One study in Nigeria explored the nuance of these interactions and noted that men were strongly influenced by their mothers and women valued the direction of fathers-in-law and elders in the community [40]. Dhaliwal et al. described how mothers-in-law use their positions as matriarch to motivate daughters-in-law to vaccinate their children; health workers then leverage this influence when making home visits [35]. Grandparents played a key role in supporting HPV immunization in South Africa [54] and influencing mothers and fathers for childhood immunization in the Philippines [44] and in Timor-Leste [55]. Both studies in South Asia noted that mothers may play a subsidiary role to maternal and paternal grandparents in decision-making which makes their support critical for childhood immunization [44,55].

Some women who opposed their husbands’ decision not to immunize their children faced **intimate partner violence** including emotional, verbal, and physical violence. While this was experienced by a minority of women in one study in Uganda, all female participants reported that they had witnessed or heard about this happening. Adolescent girls and female survivors of IPV sexual violence in Nigeria were found to have higher odds to be vaccine hesitant compared to those who did not face violence [87]. One study in South Africa also reported that child abuse was a problem in their community and women wanted the HPV vaccine for their daughters to reduce the chance of exposure to HPV if they were forced to have unprotected sex [54].

**Gendered myths and misconceptions** were also identified as barriers to immunization. Some caregivers and healthcare workers feared that different vaccinations might cause infertility for both boys and girls (n = 7). This emerged for a variety of vaccines including HPV (n = 4), COVID (n = 1), H1N1 (n = 1), and childhood vaccinations (n = 1) [18,19,20,24,30,60,70]. Another important barrier which emerged specifically for HPV was a concern that the vaccine would increase sexual activity among adolescent girls (n = 7) [18,35,54,58,71,72,73]. Similar concerns were not expressed about boys.

All studies that measured **empowerment** found a positive association between women’s empowerment and immunization coverage (n = 10) [8,9,12,13,38,65,74,88,89,90]. Empowerment was measured in different ways, usually by combining constructs of decision-making power, enabling resources, independence and agency, and attitudes towards GBV and IPV to generate a score or index. Two studies used the Survey-based Women’s emPowERment (SWPER) Global Index to measure the influence of women’s empowerment on immunization across 50–52 countries [8,9]. Wendt et al. found that the social independence domain of this index presented more consistent associations with no-DPT than other domains [8]. Johns et al. further explored social independence and found that DTP3 immunization coverage was 12.3 percentage points higher among the children of women with the highest social independence score compared with the children of women with the lowest score [9]. In other studies, multivariate models showed the intersecting role of wealth and socioeconomic status with empowerment. Because financial resources are needed to obtain vaccinations, sometimes empowerment plays a less important role when women lack the financial resources to offset the cost of vaccinating their child [13,88].

**Access** is the ability or inability to successfully carry out the transaction of vaccine utilization, representing barriers and facilitators between an individual’s intent and the health system’s readiness.

At the intersection of intent and access is women’s **occupational status** and associated norms, roles, and responsibilities. Formally employed women face additional challenges to immunizing their children such as loss of income and **opportunity costs** [25,38] and **competing priorities** or demands on their time [18,40,55,61]. Women across geographies and regardless of occupation status experienced gendered expectations of their labor resulting in increased household and caregiving demands, thus decreasing time for immunizations [23,25,40,41,55,60,62,74,75,76]. Seven studies identified a lack of male engagement in the household and in caregiving as a barrier [21,35,43,44,54,62,75] and four additional studies recommended male engagement as an important strategy given their decision-making power [19,20,63,91].

The **timing of/schedule and distance** to services can exacerbate this challenge. Distance to facilities was reported as a key barrier to immunizations services in 12 studies [17,25,28,36,61,66,67,74,78,79,80,81]. This effect was made worse for low-income women who were socially isolated [25]. Safety and security were additional concerns documented by five studies which affected women’s ability to travel and/or travel without accompaniment [25,36,57,64,92]. 

Studies reported barriers associated with **direct costs** including the cost of the vaccine [19,23,64,76,80,82,83], transportation costs to access services [19,25,29,36,58,61,68,70], and illicit fees for services or the need to pay health workers [25,36]. One woman in Malaysia reflected, “What’s the point of taking my children to a clinic to be vaccinated if I do not have money?” [28]. Seven studies also reported that women did not have decision-making power over financial resources, relying on their husbands to provide the funds and/or approve the use of funds which negatively influenced immunization [15,25,34,38,61,79,84]. Three studies found that women having their own income and discretion about spending it or joint decision-making on earning had increased odds of the children being fully immunized [16,67,74]. 

**Readiness** encompasses the health system’s supply of vaccine services to adequately meet demand. This includes immunization supplies, human resources, and related systems and structures. 

The **vaccinators** themselves, their characteristics, availability, and level of training were all important influences on immunization. One study documented a preference for more experienced health workers, which in this setting, translated to male health workers [60]. Men in the Philippines also preferred to receive the HPV vaccine from male providers, particularly physicians (not nurses) [76]. Alternatively, other studies noted that a lack of female vaccinators leads to increases in coverage inequities [10,57]. In some settings, women cannot be in contact with men who are not related to them [43,57]. Men have also expressed preferences that their daughters and wives be vaccinated by females [33,85]. In India, one study showed that districts with 50% or more Lady Medical Officers compared to those with 50% or less showed improved full immunization of children 12–23 months old (69.7% vs. 63.7%, *p* = 0.02) [85]. Many female health workers, however, face barriers to providing immunization services including feeling unsafe and being harassed by caregivers [35,57,86] and low and late remuneration [57].

The gender sensitivity of **facility** structures was also a barrier. This included lack of privacy [33], lack of gender-segregated facilities, and reduced access for transgender individuals [64]. Excessive wait times were also reported in six studies, resulting in children not receiving immunizations and/or caregivers not willing to return [25,36,60,63,66,74]. This intersects with women’s competing demands for time and availability to wait for services.

Studies described the effect of women’s prior **experiences with the healthcare system**, both positive and negative, as influencing vaccination coverage. Some women reported experiencing feelings of shame if they missed a prior appointment, forgot the child’s vaccine card, or if their children appeared malnourished or dirty [55,60,61,62,63]. Poor mothers in Uganda reported being bullied by other women and health workers; in the same setting, a women described a stigma against teenage motherhood which prevented young mothers from seeking care [38]. A study in DRC, Mozambique, and Nigeria found that caregivers who experienced disrespect in the health system were the least likely to return [25]. Members of the transgender community in Pakistan reported harassment and discrimination at COVID-19 vaccination centers which made it difficult for them to receive services during the pandemic [64]. 

Antenatal care (ANC) attendance/use emerged as an important positive predictor of immunization. This was most consistently reported in Ethiopia (n = 5) [16,66,67,68,69] where studies found that children of mothers who attended ANC were more likely to be immunized than those of mothers who did not. This was also reported in Senegal [27], Afghanistan [50], and Nepal, where children of mothers who used antenatal care were 3.31 times as likely to have received all eight vaccinations, 3.87 times as likely to have received all doses of DPT, 3.80 times as likely to have received polio vaccines, and 3.45 times as likely to have received measles vaccine [65].

Further, while few studies assessed supply-side issues, a few (n = 8) mentioned that the availability of vaccines could be an important barrier [20,36,60,61,62,63,72,74]. Schwarz et al. noted that the unavailability of vaccines leads to pessimism and future nonadherence [61]. Kagone described the harmful nature of restrictive vial opening policies which discourage health workers from opening multi-dose vials to avoid wastage; this results in delayed vaccination and increased frustration among caregivers [60]. And in one study, a respondent said, “we have become tired of this, which is why we don’t bother going there anymore” [74]. Two studies that explored availability of HPV vaccines indicated that supply could be a concern (based on prior experiences) and therefore programs and policies should prioritize only vaccinating girls and not girls and boys [20,72].

Only one study measured outcomes at the leadership level, assessing the impact of women’s political representation on child health outcomes using a dataset covering 162 countries over 30 years [93]. This study found a significant positive effect of women’s political representation on measles and DPT vaccination coverage particularly in East Asia and the Pacific, Latin America and the Caribbean, the Middle East and North Africa, South Asia, and Sub-Saharan Africa. The authors found that immunization rates are rising faster in countries with gender quota implementation.

## 4. Discussion

This review has documented a range of gender barriers to immunization, many of which are interdependent and found across geographies. What can we learn from this evidence? First, we learn that some of the most important reasons that women do not bring their children to get vaccinated lie outside of the usual purview of immunization programming. Household decision-making, for example, which is deeply entwined with social norms governing the appropriate roles for men and women within families, is often perceived as being far outside of the scope of the health system. Similarly, the fact that women often face multiple competing demands on their time is not easily addressed by immunization-focused interventions.

However, some of the gender barriers identified in the research are amenable to being addressed with adaptations to existing vaccination programming. For example, training providers on respectful patient treatment could improve women’s experiences with the healthcare system, influencing their likelihood to return for immunization services. Women’s engagement with ANC emerged as an important predictor of immunization, indicating that strengthened ANC services could help improve maternal and child immunization coverage [94]. Meeting people where they are to enhance access could include constructing more facilities or satellite clinics, providing mobile options, or conducting home visits to particularly marginalized and hesitant groups [95]. A study conducted in the urban slums of Dhaka, Bangladesh, found that extending vaccination service hours increased childhood immunization among children of employed women who could not previously attend the service window [96].

Our review found that while women are largely responsible for health care seeking, men play a critical role through their decision-making power and control of resources. This is also true for community elders and religious leaders who are able to influence men in their communities. These relationships are complex and different family members hold varying degrees of power depending on the context. But men and other family members can be better engaged throughout the immunization process to relieve the caregiving burden that women tend to manage. Equimundo recently launched their State of the World’s Fathers 2023 Report which highlights data from 17 countries showing that while fathers feel equally responsible for care work, mothers overall are still doing the majority of caregiving (3 to 7 times as much as men) [97]. But men who say they take care of their emotional selves are two to eight times more likely to care for another family member than those who do not. Given the overwhelming emotional and physical labor that women face, programs that seek to engage men, in the community and/or in places of employment, could focus on self-care and family care.

There is also abundant evidence that offering caregivers, especially those living in poverty, compensation for the direct and opportunity costs of immunization is a highly effective way of putting financial resources into the hands of women, which can enhance their economic independence and be used to vaccinate children [98,99,100]. This review documented numerous financial barriers to immunization including transportation, costs of the vaccines, giving up paid labor to attend clinics, and even the shame of appearing poor. Putting cash into women’s hands could directly ameliorate many of these gender barriers. That said, enhanced demand for immunization services must be met by supply. This review attempted to identify barriers across the immunization value chain, including on the supply side, but very few gender barriers were documented further up the chain. Only one study assessed the impact of women’s representation in leadership positions on health, including immunization [93]. However, evidence on the impact of women’s leadership on improving health outcomes is growing [101] and this could have positive implications for immunization programming, prioritization, and health systems changes.

### 4.1. Areas of Future Research

This review highlights important areas for future research on the influence of gender on global partnerships, vaccine procurement, country-led delivery, supply chains, and national vaccine markets. There is a dearth of research that can establish a causal relationship between gender barriers and vaccine outcomes. Intervention research, which seeks to evaluate interventions targeting specific gender barriers to immunization, could help address this gap, and provide much needed data on what works.

### 4.2. Strengths and Limitations

Most studies included in this review were qualitative or utilized secondary data, specifically DHS, to generate associations. It is therefore difficult to establish which, if any, gender barriers cause changes in immunization coverage. We only included articles that used an explicit gender lens in their data collection and analysis, beyond sex disaggregation, and may have missed articles that presented gender barriers. However, given the depth and breadth of articles included, and the consistency of findings among these articles, it is unlikely that additional articles would have yielded unique insights. This work was funded by the Bill and Melinda Gates Foundation Immunization Team whose portfolio focuses on select antigens, largely focusing on routine childhood vaccines. Search terms designed to meet the needs of this project may have missed vaccines outside this scope but with documented gender barriers.

### 4.3. Conclusions

This review definitively establishes that gender barriers are highly relevant in many socioeconomic contexts across LMICs. We also know that these gender barriers intersect with other widely known barriers. But women’s lack of agency over the decision to vaccinate and the ability to pay the costs to vaccinate seem to be the most common, possibly most important, factors affecting coverage. We can only know how important these barriers are with more research that measures the impact of programming on gender barriers to immunization coverage. Failing to learn about and address gender barriers to immunization is detrimental to public health programs and policies because without the generation, analysis, and synthesis of gender data, coverage will not change.

## Figures and Tables

**Figure 1 vaccines-12-00625-f001:**
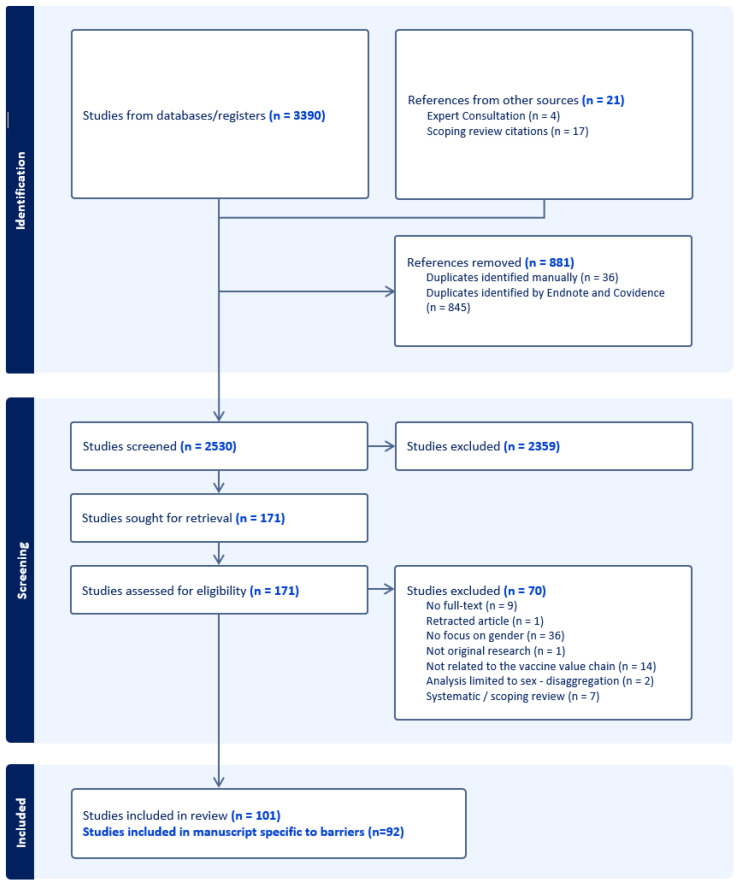
PRISMA diagram.

**Figure 2 vaccines-12-00625-f002:**
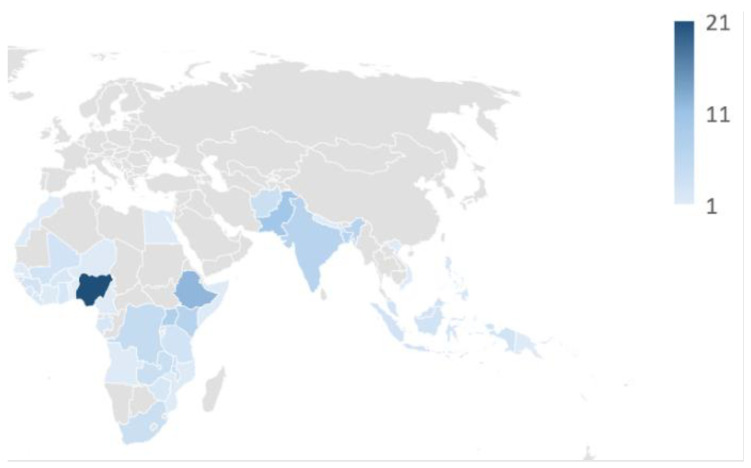
Map of documented gender drivers of immunization.

**Table 1 vaccines-12-00625-t001:** A comprehensive overview of the gender barriers documented.

Gender Influencers on Immunization	Sub-Theme	Key Points	Geographies/References
**Intent**—the demand for vaccines that would result in vaccination in the absence of other barriers.			
1. Women’s autonomous decision making	Lack of decision-making over health	-In many settings women rely on their husband’s or an elder’s permission to seek healthcare services including immunization. -Women with high household decision-making are more likely to have fully immunized children. -Women who make decision jointly with their husbands are more likely to have fully immunized children than when husbands make decision alone. -Some women who oppose their husband’s decision face increased risk of intimate partner violence.	Africa (n = 37) South Asia (n = 16) Cross-country (n = 3) [8,12,17,18,19,20,21,22,23,24,25,26,27,28,29,30,31,32,33,34,35,36,37,38,39,40,41,42,43,44,45,46,47,48,49,50,51,52,53,54,55,56,57,58,59]
2. Past experiences with the health system	Negative Experiences	-Some women reported being shamed/bullied by health workers if they missed a prior appointment, forgot the child’s vaccine card, or if they or their child appeared dirty and/or malnourished. -Caregivers were not always provided with complete information about the vaccination, including likely side effects and how best to alleviate those side effects. -Caregivers who experience disrespectful treatment are least likely to return to the health system.	Burkina Faso, DRC, Ethiopia Gabon, Nigeria, Mozambique, Pakistan, Timor-Leste, Uganda (n = 8) [25,38,55,60,61,62,63,64]
	Engagement with ANC	Women who attend ANC are more likely to have fully immunized children than women who do not.	Afghanistan, Ethiopia, Nepal, Senegal (n = 8) [16,27,50,65,66,67,68,69]
3. Gendered myths, misconceptions	Fears of infertility	-Caregivers and health workers expressed concerns that vaccines (including HPV, COVID-19, H1N1, and childhood immunizations) could cause infertility.	Burkina Faso, Kenya, Malawi, Morocco, Tanzania, Zambia (n = 7) [18,19,20,24,30,60,70]
	Promotion of earlier/increased sexual activity	-Caregivers feared that HPV immunization would result in earlier sexual debut/increased sexual activity for adolescent girls.	Ethiopia, Ghana, India, Malawi, Papua New Guinea, South Africa, Zimbabwe [18,35,54,58,71,72,73]
**Access**—ability or inability to successful carry out the transaction of vaccine utilization.			
1. Time poverty	Competing and gendered demands on time	-Women face competing demands on their time including employment and gendered expectations of caregiving and household labor. This reduces their time for immunizations. -Men’s limited contribution in unpaid domestic work exacerbates the demand on women’s time.	Burkina Faso, Ethiopia, Nigeria, DRC, Mozambique, Sierra Leone, South Africa, India, Philippines, Somalia, Malaysia, Philippines, Pakistan, Timor-Leste, Uganda, Malawi, Gabon [18,21,23,25,35,40,41,43,44,54,55,60,61,62,74,75,76,77]
	Distance to facilities	-Timing of (schedule) and distance to services can exacerbate this challenge. This effect is worse for low-income women who are socially isolated.	Gabon, Malawi, Nigeria, Malaysia, South Africa, Uganda, Guinea, Malawi, Ethiopia, DRC, Mozambique, Bangladesh [17,25,28,36,61,66,67,74,78,79,80,81].
2. Direct costs	Costs of vaccines, transportation to services, and illicit fees.	-Women reported that the cost of vaccine, transportation costs to access facilities, and illicit fees for services were barriers to immunization.	DRC, Ethiopia, Gabon, Guinea, India, Kenya, Malaysia, Mozambique, Nigeria, Pakistan, Philippines, South Africa, Uganda, Zambia, Zimbabwe [19,23,25,28,29,36,58,61,64,68,70,76,80,82,83]
	Financial Agency	-Women lack financial agency, relying on their husbands to provide the funds and/or approve use of funds for immunization. -Women with their own income and discretion about spending it had increased odds of their children being fully immunized.	DRC, Ethiopia, Gabon, Mozambique, India, Nigeria, Uganda [15,16,25,34,38,61,67,74,79,84]
**Readiness**—encompasses the health system’s supply of vaccine services to adequately meet demand.			
1. Vaccinators/Health care providers	Lack of women vaccinators/preference for women vaccinators	-A lack of women vaccinators leads to increased coverage inequities, and many men prefer women/daughters are vaccinated by women.	Bangladesh, DRC, Ethiopia, India, Nigeria, Pakistan, Somalia [10,33,43,57,85]
	Women workers’ occupational concerns	-Many women health workers experience safety issues, harassment, and low or late remuneration for their services.	Afghanistan, Bangladesh, DRC, Ethiopia, India, Nigeria, [35,86]
2. Health care facilities	Gender unintentional facilities	-Lack of privacy and gender-responsive facilities (i.e., functional and separate washrooms and security for transgender individuals) is a barrier.	Bangladesh, Pakistan [33,64]
	Excessive wait times	-Excessive wait times result in children not receiving immunizations and/or caregivers not being willing to return.	Burkina Faso, DRC, Guinea, Ethiopia, Mozambique, Nigeria, Uganda [25,36,60,63,66,74].
3. Vaccine availability	Vaccine stockouts	-Unavailability of vaccines can lead to pessimism and future nonadherence. -Restrictive vial opening policies result in delayed vaccination and increased frustration among caregivers.	Burkina Faso, Ethiopia, Gabon, Guinea, Nigeria, Papua New Guinea, Tanzania, Uganda [20,36,60,61,62,63,72,74].

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
