# Peer review of "Of Money and Men: A Scoping Review to Map Gender Barriers to Immunization Coverage in Low- and Middle-Income Countries"

_vaccines, 2024, doi:10.3390/vaccines12060625_

Round 1

Reviewer 1 Report

Comments and Suggestions for Authors

Thank you for sharing the paper with me for review. The paper is well-written, well-organized, and has the potential to make a substantial contribution to the scientific literature. Despit the obvious strengths of the paper, I do have a few reservations limiting my enthusiasm. My comments regarding strengths and weaknesses are provided below:

Strengths

(1) Well-written, grammatically and structurally sound.  The information flows nicely throughout all sections of the paper.

(2) Good documentation of the PRISMA guidelines, easily, well articulated flow of how the sample was derived. 

(3) Good use of multiple data-bases, the study team covered the main ones with access the relevant periodicals.

(4) Very nice link between the conceptual foundation reported on page 2 of 12 and the way the findings were provided. 

Areas of Concern

(1) Articulation of qualitative design leading to the structure of the interpretive text used to support the numerical findings is nonexistent. In other words, what was the analytical process that yielded the information presented here?

(2) Broadly, it is not sufficient to say that the "data were analyzed in Microsoft Excel." There should be meaning to the section titled Methods.  The  analytical strategies should be as well described as the steps used to find the relevant articles. The paper is silent in this regard and thus not reproducible in its current form.  

(3) The suporting table is informative, which provides the evidence for the interpretive statements, is not very large for one that holds the information from 92 published scientific papers. I would have expected a much more developed table.  The authors should either include this or explain how this is the case.  

(4) Discussion of direct costs telegraphed in the abstract was harder to find in the Results and Discusion sections (which was different from the other major categories).  Perhaps this could be more clearly conveyed in subsequent drafts of the paper. 

Author Response

Thank you for your thoughtful comments. Below we have included a point by point response to your concerns. 

1) Articulation of qualitative design leading to the structure of the interpretive text used to support the numerical findings is nonexistent. In other words, what was the analytical process that yielded the information presented here?

Thank you for raising this concern. We have added more detail to the methods section to address the concern and the second concern which also emphasizes the initial lack of detail in the methods section. Specifically we note, "Descriptive statistics were generated where data could be quantified. Data was initially charted along the immunization value chain, using a gender analysis matrix. This approach seeks to organize data along gender analysis domains (i.e. access to resources; distribution of labor, practices, and roles; norms, values, and beliefs; decision-making power and autonomy; and policies, laws, and institutions. Results largely converged in select domains, requiring and additional lens." We then go on to describe Phillips et al's conceptual framework which subsequently informed analysis and reporting. 

(2) Broadly, it is not sufficient to say that the "data were analyzed in Microsoft Excel." There should be meaning to the section titled Methods.  The  analytical strategies should be as well described as the steps used to find the relevant articles. The paper is silent in this regard and thus not reproducible in its current form.  

(3) The suporting table is informative, which provides the evidence for the interpretive statements, is not very large for one that holds the information from 92 published scientific papers. I would have expected a much more developed table.  The authors should either include this or explain how this is the case.  

Given the number of articles included in this review we felt there was an advantage in conveying the main findings in a consolidated format rather than providing data on each individual article. 

(4) Discussion of direct costs telegraphed in the abstract was harder to find in the Results and Discusion sections (which was different from the other major categories).  Perhaps this could be more clearly conveyed in subsequent drafts of the paper. 

Given the interconnectedness between direct costs and other issues of intent and access, we understand the reviewer's concern that this may be less distinct and perhaps harder to find. We have ensured that 'direct costs' are bolded in the results (see line 204). Further, the entire final paragraph prior to 'areas of future research', starting on line 308, is dedicated to exploration of direct costs and the importance of putting cash into women's hands. 

Reviewer 2 Report

Comments and Suggestions for Authors

This review article points out that genders are the most universal factors that impede maternal and childhood immunization in low- and middle-income countries. The author proposed a novel angle to point out the reasons and the problems that need to be paid attention to. The authors propose that the economic status of women is one of the main reasons that result in the immunization barriers. The authors applied three databases, PubMed, Embase, and CINAHL to collect the published data focusing on gender, immunization, geography. In addition, the availability of vaccine is one of the main reasons that lead to the low percentage of immunization coverage. The authors discussed HPV, COVID-19, Polio, maternal vaccines, yellow fever and H1N1 across 43 countries in Africa and south Asia. But the number of cases is limited in each vaccine that the author has been discussed. Furthermore, it will be useful if the authors can take a look at other types of vaccines as well since the vaccine such as COVID-19 is very special mRNA vaccine which will lose functions in a year and need to be injected annually or even a short period that depends on different situations. For the children or infants, vaccines such as cholera, MMR, varicella, BCG, measles, Japanese encephalitis, DTP, may be more useful.

Author Response

Thank you for your thoughtful comments on this manuscript. We chose to focus on childhood vaccinations and HPV because the original work was commissioned by the BMGF Immunization Team whose portfolio focuses on these antigens. We have added a note about this in the introduction and in the limitations sections of the manuscript. 

Round 2

Reviewer 1 Report

Comments and Suggestions for Authors

Thanks to the authors for their responses to my questions and concerns. The paper is fairly strong in all respects except for the study design components.  It is not at all clear how the data were mapped and verified to the points presented in text and in the supplementary table (not, in my opinion, reproducible).  I will defer to the editor to determine if this meets the reporting standards of the journal.  

Comments on the Quality of English Language

Acceptable.  Proof-reading of the revsied text would have been helpful. 

Author Response

Dear reviewer, thank you for continuing to raise this concern and to push the team to better enumerate the analytic processes we used in this review. We have added (and proof read) text to the analysis section to better describe our process.